# Study on the Structural Changes of Boneless Chicken Claw Collagen and Its Effect on Water Retention Performance

**DOI:** 10.3390/foods13223682

**Published:** 2024-11-19

**Authors:** Zheng Tang, Yiguo He, Jing Zhang, Zhifeng Zhao, Yiming Nie, Xingxiu Zhao

**Affiliations:** College of Bioengineering, Sichuan University of Science and Engineering, Yibin 644000, China; 323083202113@stu.suse.edu.cn (Z.T.); heyiguo@suse.edu.cn (Y.H.); 13880046872@163.com (J.Z.); zhaozhifeng_scu@foxmail.com (Z.Z.); 323095102315@stu.suse.edu.cn (Y.N.)

**Keywords:** chicken claw, collagen swelling, water retention, structure analysis, SEM, UV, FTIR, CD, surface hydrophobicity, LF-NMR

## Abstract

The purpose of this study was to explore the water retention mechanism of chicken claws by detecting the structural changes in collagen in boneless chicken claws under different expansion rates. Firstly, boneless chicken claw collagen with different expansion rates (0%, 10%, 20%, 30%, 40%, 50%) was extracted by the acid–enzyme complex method, and the changes in collagen were determined by scanning electron microscopy (SEM), ultraviolet spectroscopy (UV), Fourier transform infrared spectroscopy (FTIR), circular dichroism (CD), low-field nuclear magnetic resonance LF-NMR) and surface hydrophobicity to explore the mechanism that leads to changes in the water retention performance. The results of scanning electron microscopy showed that with the increase in the expansion rate, collagen molecules showed curling, shrinking, breaking and crosslinking, forming a loose and irregular pore-like denatured collagen structure. UV analysis showed that the maximum absorption wavelength of chicken claw collagen was blue shifted under different expansion rates, and the maximum absorption peak intensity increased first and then decreased with the increase in expansion rate. The FTIR results showed that collagen had obvious characteristic absorption peaks in the amide A, B, I, II and III regions under different expansion rates, and that the intensity and position of the characteristic absorption peaks changed with the expansion rate. The results of the CD analysis showed that collagen at different expansion rates had obvious positive absorption peaks at 222 nm, and that the position of negative absorption peaks was red shifted with the increase in expansion rate. This shows that the expansion treatment makes the collagen of chicken claw partially denatured, and that the triple helix structure becomes relaxed or unwound, which provides more space for the combination of water molecules, thus enhancing the water absorption capacity of boneless chicken claw. The results of the surface hydrophobicity test showed that the surface hydrophobicity of boneless chicken claw collagen increased with the increase in expansion rate and reached the maximum at a 30% expansion rate, and then decreased with the further increase in the expansion rate. The results of LF-NMR showed that the water content of boneless chicken claws increased significantly after the expansion treatment, and that the water retention performance of chicken claws was further enhanced with the increase in the expansion rate. In this study, boneless chicken claws were used as raw materials, and the expansion process of boneless chicken claws was optimized by acid combined with a water-retaining agent, which improved the expansion rate of boneless chicken claws and the quality of boneless chicken claws. The effects of the swelling degree on the collagen structure, water absorption and water retention properties of boneless chicken claws were revealed by structural characterization. These findings explain the changes in the water retention of boneless chicken claws after expansion. By optimizing the expansion treatment process, the water retention performance and market added value of chicken feet products can be significantly improved, which is of great economic significance.

## 1. Introduction

China is one of the world’s most populous countries, leading to a tremendous demand for food. In the Chinese diet, poultry is the second most consumed type of meat [1]. Chicken is the most important component of poultry meat. According to the statistics of the National Bureau of Statistics of China, chicken production accounted for 63% of poultry meat production in 2020, accounting for about 19.2% of total edible meat, second only to pork.

As one of the world’s top four broiler producers, chicken farming is extremely common in China. Its short breeding cycle and high return on investment have made significant contributions to China’s food security. Especially after the outbreak of African swine fever in 2018, which drastically reduced pork production, chicken played a critical role in filling the gap in pork supply [2].

Chicken claws, a byproduct of chicken processing, contain skin, bone, muscle, and collagen. In some Western countries, such as those in Europe and the United States, chicken claws are considered waste and not utilized. Major chicken-producing countries like Brazil and the United States, with annual chicken production reaching millions of tons, often export large quantities of chicken byproducts, including chicken claws, to other countries at low prices, including China. According to China’s General Administration of Customs, in the first 11 months of 2023, China imported 491,000 tons of frozen chicken claws. The tradition of consuming chicken claws in China dates back centuries, particularly in the southern regions, where they are processed into various specialty foods. Therefore, the deep processing of low-cost chicken byproducts into high-value chicken claw products can significantly enhance economic benefits.

In Chinese cuisine, chicken claws can be used in soups, braises, and stews, or cooked and pickled with other ingredients. For example, pickled chicken claws with chili, popular in the Sichuan and Chongqing regions, are known for their spicy, tangy flavor and are highly appetizing. In Guangdong, chicken claws are primarily processed into “Baiyun” chicken claws and “Tiger Skin” chicken claws, frequently appearing in dim sum and banquets. Initially, these chicken claw products are mainly produced by restaurants and small workshops, involving complex processing techniques and requiring the removal of bones before consumption. With changing dietary preferences, boneless chicken claws, which are convenient to eat without location or environmental constraints, have become increasingly popular among consumers.

International researchers have primarily focused on extracting and utilizing collagen from chicken claws due to their high collagen content, showcasing its wide range of applications in food, medicine, material science, and the transformation of waste natural resources into high-quality products [3,4]. HASHEM M S [4] et al. developed a new type of sulfonated gelatin from chicken claws that exhibited good antibacterial and antioxidant activity after graft polymerization. SANTANA J C C [5] et al. extracted collagen from “waste” chicken claws to produce gelatin and bioplastics. RATNA [6] et al. further transformed chicken claws into edible gelatin films and studied their properties at different concentrations to improve the quality of edible film packaging for chicken claws. SOMPIEM [7] et al. prepared edible film solutions from chicken claw gelatin to soak chicken meat, significantly improving the meat’s water retention and quality. Additionally, gelatin from chicken claws can replace certain ingredients in food to enhance product quality; for example, ARAúJO Í B S [8] et al. used chicken claw gelatin to replace fat in cooked chicken sausages, positively affecting the sausages’ quality.

At present, some studies have explored the collagen extracted from chicken claws to varying degrees, and it can be seen from these studies that collagen is closely related to the water-holding capacity of chicken. However, the water retention performance of chicken claws after expansion has not been explored. For the type of chicken claws, the taste and texture of chicken claws are some of the criteria used to judge the quality of chicken claws, and water retention affects the taste and texture of chicken claws. Therefore, the water retention of chicken claws after expansion is a very important indicator. However, there are few studies on chicken claws. The research in this paper mainly aims to explore the changes in the structure of collagen at all levels after the expansion of chicken claws and the effects of these changes on the water retention performance of chicken claws. The research in this paper can provide some theoretical basis for further research on chicken claw expansion and other undiscovered functions of collagen in chicken claws.

## 2. Materials and Methods

### 2.1. Materials

Frozen boneless chicken claws from Yibin, China.

#### 2.1.1. Experimental Reagents

Analytically pure N-butanol and glacial acetic acid were obtained from Chengdu Cologne Chemicals Co., Ltd. (Chengdu, China). Analytically pure pepsin, sodium chloride, potassium bromide and bromophenol blue were obtained from Sinopharm Group Chemical Reagents Co., Ltd. (Shanghai, China).

#### 2.1.2. Instruments and Equipment

The ordinary balance model used was HJ-50002, from Hangzhou Youheng Double Co., Ltd. (Hangzhou, China). The electronic analytical balance model used was AX124ZH/E, from Ohaus Instruments Co., Ltd. (Changzhou, China). The constant temperature water bath model used was B 260, from Shanghai Yarong Biochemical Instrument Factory (Shanghai, China). The vacuum refrigerated centrifuge model used was RTI6000 C, from Shanghai Rongwei Instrument Co., Ltd. (Shanghai, China). The vacuum freeze dryer model used was Universal 32 R, from Dalian Jielun Drying Equipment Co., Ltd. (Dalian, China). The vortex oscillator model used was SCI-VS, from Shandong Boke Biological Industry Co., Ltd. (Jinan, China). The UV spectrophotometer model used was UV-1601, from Fuyue Biotechnology (Shanghai) Co., Ltd. (Shanghai, China). The Fourier transform infrared spectrometer model used was AF-2900 PLUS, from Opus Instruments Group (Zhengzhou, China), and the circular dichroism spectrometer model used was Chirascan VX, from the Shanghai representative office of the British Applied Photophysics Company (Shanghai, China). The low-field nuclear magnetic resonance model used was NMI20-040V-1, SF = 21 MHz, from Suzhou Newmai Instrument Co., Ltd. (Suzhou, China). The scanning electron microscope model used was VEGA 3SBU, from Aoyi Instrument Co., Ltd. (Shanghai, China).

### 2.2. Experimental Methods

#### 2.2.1. Preparation of Chicken Claws Under Different Expansion Rates

In the expansion process, boneless chicken claws were expanded with a glacial acetic acid concentration of 0.9%, sorbitol addition of 0.15%, sodium hexametaphosphate addition of 0.006%, sodium citrate addition of 0.005% and sodium dihydrogen phosphate addition of 0.005% (without flavoring treatment). Boneless chicken claws with different expansion rates (0%, 10%, 20%, 30%, 40%, 50%) were prepared by controlling the expansion time.

#### 2.2.2. Extraction of Collagen from Boneless Chicken Claws

We referred to the method of Zheng Qingyao [9] et al. to extract sea cucumber collagen, and modified it according to this method. The chicken claws under different expansion rates were taken, the impurities were removed, and the chicken claw skin was chopped. According to Figure 1, 10% n-butanol was added at 1:10 (*w*/*v*) to soak for 24 h to remove residual fat and clean it. A 0.5 mol/L acetic acid solution containing 0.1% pepsin was added at a ratio of 1:20 (*w*/*v*) and extracted for 48 h. The supernatant was centrifuged and NaCl was added to 2.4 mol/L to salt out the collagen. After centrifugation, the precipitate was dissolved in 0.5 mol/L acetic acid as little as possible, and then dialyzed in 0.1 mol/L acetic acid solution for 24 h and distilled water for 48 h. The dialysate was changed every 12 h. After vacuum freeze-drying, it was stored at 4 °C.

#### 2.2.3. Determination of Scanning Electron Microscopy

The collagen extracted from the expanded boneless chicken claws was freeze-dried, and the boneless chicken claw collagen under different expansion rates (10%, 20%, 30%, 40%, 50%) after freeze-drying was fixed on the metal sample table with conductive adhesive. After gold spraying, the microstructure of collagen with different expansion rates was observed using a scanning electron microscope (VEGA 3SBU, Aoyi Instrument Co., Ltd., Shanghai, China).

#### 2.2.4. Determination of UV Spectrum

The boneless chicken claw collagen with different expansion rates after freeze-drying was dissolved in 0.1 mol/L glacial acetic acid to prepare a 0.5 mg/mL collagen solution. With 0.1 mol/L glacial acetic acid solution as a blank reference, the collagen solution at different expansion rates was scanned by an ultraviolet spectrophotometer (UV-1601) at 190–400 nm [10].

#### 2.2.5. Determination by Fourier Infrared Spectroscopy

The freeze-dried collagen was mixed with dried KBr at a ratio of 1:100 and uniformly ground into powder. The full-wavelength scanning was performed using an infrared spectrometer (AF-2900PLUS) at 400~4000 cm^−1^ [11].

#### 2.2.6. Determination of Circular Dichroism

The freeze-dried collagen was prepared into a 0.5 mg/mL collagen solution with 0.1 mo/L acetic acid solution, and scanned with a 1 mm quartz cuvette in the range of 190–260 nm of a circular dichroism spectrometer (Chirascan VX) [12].

#### 2.2.7. Determination of Surface Hydrophobicity

We referred to the method of Chin et al. [13] and made some modifications. The freeze-dried collagen was prepared into a 1.0 mol/L collagen solution with a 0.1 mol/L acetic acid solution. A total of 200 μL bromophenol blue solution (1.0 mg/mL) was added to 1 mL of collagen solution, and vortexed for 10 min. After centrifugation (6000× *g*, 15 min), 0.4 mL of supernatant and 3.6 mL of acetic acid solution were mixed to determine the absorbance value A sample at 595 nm. The absorbance value A blank of 0.1 mol/L acetic acid under the same conditions was used as the blank control, and distilled water was used as the blank reference. The surface hydrophobicity of collagen at different expansion rates is expressed as the content of bound bromophenol blue, calculated using the following formula:(1)$$bromophenol\;blue\left(\mu g\right)=\frac{A_{blank}-A_{specimen}}{A_{\mathrm{blank}}}\text{×}200$$

#### 2.2.8. LF-NMR Moisture Migration Determination

The boneless chicken claws (0~50%) with different expansion degrees were measured by low-field nuclear magnetic resonance (NMI20-040V-1, SF = 21 MHz) to determine the water migration in the time period of 0~1000 ms.

#### 2.2.9. Analysis of Data

In this study, the data were obtained by three parallel independent sample determinations, and the data processing software SPSS 26.0 was used for statistics and analysis. The data were analyzed by one-way ANOVA. *p* < 0.05 was considered to be significantly different, and different letters indicated significant differences between the data. Origin (2021b) was used for drawing processing.

## 3. Results and Discussion

### 3.1. Microstructure Analysis

Figure 2 is the scanning electron microscope structure of chicken claw collagen under different expansion rates. From the diagram, it can be seen that compared with the non-expanded group, the microstructure of the chicken claw collagen treated by expansion has changed significantly. The collagen under the expansion rate of 0% showed a porous structure, with a thin mesh feature and several layers of different thicknesses that form a three-dimensional network structure, with a large pore size and network structure. With the deepening of the expansion degree, the collagen at a 10% expansion rate formed a complex porous network structure with multi-level network characteristics. There are more connections and intersections, and the edges are obviously curled and contracted, so that the hydrophobic groups inside the collagen molecules are exposed, and the surface hydrophobicity of the collagen molecules is increased. At this time, the hydrophobic interaction can promote combination between water molecules and collagen molecules, which enables boneless chicken claws to absorb more water. Collagen curled at a 20% expansion rate, the degree of shrinkage deepened, and a three-dimensional network structure with more abundant sense of hierarchy was formed. Some collagen ruptured and the pores slightly enlarged. When the expansion rate reached 30%, the degree of collagen breakage and fragmentation increased, and it further curled, contracted, and cross-linked. Some large collagen molecules were broken into small collagen molecules, and new connections were formed between different collagen molecules. At the same time, the sample was still pure in a large number of connections and intersections, with a multi-level network structure. When the degree of expansion reached 40%, the degree of fragmentation in some collagen became deeper and was still pure in the connection and cross, with a multi-level three-dimensional network structure, but it became relatively large; when the expansion rate reached 50%, the degree of denaturation of some collagen deepened, which in turn deepened the degree of fragmentation and agglomeration, forming a loose and irregular pore-like degenerated collagen structure, which provided more space and channels for the entry of water molecules. Moisture filled the pores of the collagen structure to form a stable hydration layer, which effectively locked water inside the chicken claw and meant that it not easy to lose.

### 3.2. Spectroscopy

There are C=O, -COOH and-CO-NH-functional groups in the polypeptide chain of collagen, which makes it have a significant absorption peak at a wavelength of about 230 nm [14]. According to the UV absorption spectrum of collagen, it can not only determine whether there are amino acids with a conjugated π-bond chromophore such as tyrosine and tryptophan in collagen, but also determine the integrity of non-helical terminal peptides in the polypeptide chain [15]. From Figure 3, it can be seen that there is no obvious absorption peak near a wavelength of 280 nm in the chicken claw collagen under different expansion rates, indicating that the content of aromatic amino acids with a conjugated π bond chromophore in the extracted collagen product is extremely low, which is consistent with the amino acid composition characteristics of type I collagen [16]. In the wavelength range of 220–230 nm, because the collagen peptide chain contains chromophore groups such as -C=O, -COOH and CO-NH_2_, and because it is rich in hydroxyproline, proline and glycine, it will produce a strong absorption peak at 230 nm [17]. The maximum absorption wavelength of collagen changed under different expansion rates. The maximum absorption wavelength of collagen without expansion treatment appeared at 228 nm, which was consistent with the ultraviolet absorption of type I collagen. With the increase in the expansion rate, the maximum absorption wavelengths of the 10%, 20%, 30%, 40% and 50% treatment groups were blue shifted to 225, 227, 228, 225 and 225 nm, respectively, and the maximum absorption peak intensity increased first and then decreased with the increase in the expansion rate. This shows that with the increase in the expansion rate, the content of hydrogen bonds and ionic bonds in collagen increases, which causes the conformation of collagen to change, and the external environment of the collagen skeleton also changes, resulting in the movement of the absorption peak position of boneless chicken claw collagen [18].

### 3.3. Infrared Spectrum

It can be seen from Figure 4 that the collagen of chicken claws under different expansion rates has obvious characteristic absorption peaks in the amide A, B, I, II and III regions, which is consistent with the typical type I collagen in the infrared absorption peak position [19]. The characteristic peak absorption frequency of collagen under different expansion rates is not much different, but the characteristic absorption peak intensity is different. The different absorption spectra also indicate that the secondary structure of chicken claw collagen changes due to different degrees of expansion. The characteristic absorption peak of the amide A band is caused by the stretching vibration of N-H, which usually appears at about 3400~3440 cm^−1^ [20], but when the N-H bond is involved in the formation of a hydrogen bond, the characteristic absorption peak will blue shift to about 3300 cm^−1^ [21]. From Table 1, it can be seen that compared with a 0% expansion rate, hydrogen bonds are formed in collagen after acid expansion, so that boneless chicken claws can absorb and retain more water; in addition, with the increase in the degree of expansion, the position of the maximum absorption wavelength of the amide A band showed a dynamic change, which may be due to the change in the number of hydrogen bonds inside the collagen after the expansion of the boneless chicken claw, and the hydrogen bond interaction between the collagen molecules changed [22]. When the expansion rate is 50%, the amide A band is located at 3325 cm^−1^, indicating that compared with other groups, the 50% expansion rate group contains more hydrogen bonds, which is more conducive to the absorption and binding of water, which is consistent with Jia Yuanjun‘s [23] research.

The amide B band is related to the asymmetric stretching vibration of CH_2_, and the characteristic absorption peak is usually between 2924 and 2932 cm^−1^ [24]. The characteristic absorption peaks of the collagen amide B bands at different expansion rates were in the range of 2924~2928 cm^−1^, indicating that there were the asymmetric stretching vibrations of CH_2_ in collagen at different expansion rates. With the increase in expansion rate, the vibration frequency of the CH_2_ bond changed, and the maximum absorption wavelength of each group of amide B bands was red shifted and then blue shifted. The hydrophobic interaction between collagen molecules changed, which enabled the absorption of more water by boneless chicken claws.

The characteristic absorption peak of the amide I band is in the range of 1600~1700 cm^−1^, which is related to the stretching vibration of C=O. When the absorption peak wave number is larger, the peptide chain structure is more orderly. Compared with the non-expansion group, the maximum absorption wavelength of the amide I band of the different expansion rate treatment groups showed different degrees of blue shift, indicating that the entry of water molecules and the weakening of the interaction between molecules after expansion reduced the overall order of collagen, but made it easier to absorb collagen and retain water, which was consistent with the study of Liu Fangfang [25] et al.

The amide II band with the characteristic absorption peak in the range of 1500~1600 cm^−1^ is caused by the stretching vibration of C-N and the plane bending vibration of N-H [26]. With the increase in expansion rate, the collagen amide II band showed a phenomenon of red shift first and then blue shift. In the early stage of expansion, water molecules interact with collagen molecules to form hydrogen bonds. With the destruction of the original hydrogen bond network, the absorption peak wavelength of the amide II band is red shifted, and the hydrophilicity of the collagen surface is enhanced, which is conducive to the absorption of more water by boneless chicken claws. However, the absorption peak position of collagen did not change at a 10–30% expansion rate, indicating that the changes in the C-N bond in the collagen subunit structure tended to be stable. As the expansion rate continues to increase, the hydrogen bond network within the collagen molecule is further adjusted or reorganized, causing the blue shift of the absorption peak wavelength of the amide II band, and forming a more stable structure that is conducive to the entry of water molecules, which provides collagen with a greater water absorption capacity and the ability to accommodate more water molecules.

The amide III band with a characteristic absorption peak in the range of 1200~1300 cm^−1^ can reflect the characteristics of collagen and participate in the formation of a triple helix structure. It has a variety of vibration modes, usually N-H bending vibration [19]. With the deepening of the expansion degree, the maximum absorption wavelength of the amide III band first blue shifted, then red shifted, and then blue shifted. The reason is that with the increase in the expansion rate, the vibration frequency of the N-H bond increases, and the collagen molecules are locally ordered at the initial stage of water absorption. The blue shift of the absorption peak wavelength helps the collagen molecules to form a stable hydration layer; during the further expansion process, the vibration frequency of the N-H bond decreases, and the collagen structure becomes relaxed and disordered, thereby providing more water absorption space. At a higher expansion rate, the vibration state of the N-H bond undergoes a new adjustment to form a new ordered structure, so that the boneless chicken claws still maintain a certain water retention performance under a high expansion state.

### 3.4. Circular Dichroism Analysis

It can be seen from Figure 5 that collagen showed a significant positive absorption peak at 222 nm under different expansion rates, and that the positive and negative absorption turning points appeared at 215 nm, while the negative absorption peaks appeared at 199 nm, 201 nm, 202 nm, 201 nm, 200 nm and 203 nm, respectively. With the increase in expansion, the negative absorption peak of chicken claw collagen has different degrees of red shift, indicating that the expansion treatment makes the chicken claw collagen partially denatured or part of the triple helix structure become more relaxed or untwisted, providing more binding space and channels for water molecules, thereby enhancing the water absorption capacity of boneless chicken claws. When the expansion rate reached 50%, the degree of collagen denaturation was more serious, but the positive absorption peak at 222 nm did not disappear, indicating that the collagen in boneless chicken claws was not completely denatured at this time, and the triple helix structure was not completely destroyed. Boneless chicken claws have certain water retention properties, and can enhance water absorption. The positive and negative peak intensity ratios (Rpn) of the collagen circular dichroism spectra at different expansion rates were calculated to be 0.10, 0.13, 0.14, 0.13, 0.12, and 0.14, respectively, all in the range of 0.1–0.15, indicating that the chicken claw collagen still retained the triple helix structure after different degrees of expansion treatment [27]. Therefore, the circular dichroism spectrum of collagen still conforms to the characteristic spectrum of a collagen triple helix structure [28].

### 3.5. Surface Hydrophobicity Analysis

It can be seen from Figure 6 that with the increase in the expansion rate, the surface hydrophobicity of boneless chicken claw collagen increased, reached the maximum at 30%, and then decreased with the increase in the expansion rate. With an expansion rate in the range of 0–30%, the surface hydrophobicity of collagen showed a significant increasing trend. The results showed that with the increase in the expansion degree, the tertiary structure of boneless chicken claw collagen changed continuously. In this process, the force to maintain the structure of collagen gradually weakened, resulting in the loosening of its triple helix structure and the degeneration of collagen. As the collagen molecules are broken and broken, the hydrophobic groups are gradually exposed, so that the surface hydrophobicity increases continuously, and the interaction between the hydrophobic groups and the water molecules helps the boneless chicken claws to absorb water more effectively [29]. When the expansion rate reaches 30%, the degree of collagen denaturation increases, the broken collagen molecules aggregate, some hydrophobic groups are covered by hydrophilic groups or form hydrogen bonds with water molecules, and the surface hydrophobicity decreases [30]. The hydrogen bond interaction helps boneless chicken claws to maintain the absorbed water more effectively.

### 3.6. LF-NMR Analysis

The transverse relaxation time (T_2_) is also known as the spin–spin relaxation time [31]. The T_2_ inversion spectra of different expansion rates can be obtained by LF-NMR technology. The smaller the T_2_, the smaller the degree of freedom of the proton, the tighter the combination of water and matter, and the more difficult it is to remove when it is dry; the greater the T_2_, the greater the degree of freedom of the proton, the greater the fluidity of the water, and the easier it is to discharge during the drying process [32]. From Figure 7, it can be concluded that there are two different states of moisture in boneless chicken claws with different expansion rates: T_21_ (<18 ms) and T_22_ (18~1000 ms). Among them, T_21_ has the shortest relaxation time at this stage, which is the bound water distributed in chicken claws. This part of water is closely combined with the collagen in chicken claws through hydrogen bonds [33]. The binding force is strong, the fluidity is poor, and Moisture is difficult to separate [34]. The free water T_22_ peak is the highest peak, which exists in protoplasm, the collagen network structure and the intercellular space, with a high degree of freedom and good fluidity [34]. Further analysis shows that the water migration of boneless chicken claws under different expansion rates can be divided into three stages: the initial stage, middle stage and late stage. In the initial stage (<18 ms), the peak area ratio of each curve of each expanded boneless chicken claw was close to zero, and the water content of each concentration sample was almost kept at a low level. At this time, the water is mainly bound water, which is tightly bound to the protein and other macromolecular structures of chicken claws, and is not easily detected or migrated. Bound water refers to water that is tightly bound to macromolecules such as proteins and lipids through weak interaction forces such as hydrogen bonds and van der Waals forces [33]. Because of its close combination with the molecular structure of chicken claws, the migration speed is slow, showing strong water retention performance. In the middle stage (18~100 ms), the moisture content of each concentration sample began to increase rapidly, and the curve showed an obvious upward trend. At this stage, part of the bound water begins to transform into transitional water. Transition water is a kind of water between bound water and free water. It is not so closely combined with the molecular structure of chicken claws, but it is not completely free water. This kind of water may be combined with cell structure, collagen and other components to a certain extent, but it can migrate relatively easily when external conditions (such as temperature, pressure, etc.) change. In the later stage (100~1000 ms), the water content of each concentration sample began to decrease after reaching the peak, indicating that free water existed in large quantities in the system and gradually migrated or was lost. Free water refers to the water that is not tightly bound to the macromolecular structure and can flow freely in the tissue [34]. Because of its weak combination with the structure of chicken claws, it easily migrates under the influence of the external environment.

## 4. Conclusions

In this paper, the structural changes in collagen during water swelling and its effect on the water absorption and water retention properties of boneless chicken claws were studied. The collagen of boneless chicken claws with different expansion rates was extracted by the acid-– complex method. The relationship between the structural changes in collagen and the water absorption and water retention of boneless chicken claws with different expansion rates (0%, 10%, 20%, 30%, 40%, 50%) was investigated by means of scanning electron microscopy, ultraviolet spectroscopy, infrared spectroscopy, circular dichroism and surface hydrophobicity. The results are as follows: Scanning electron microscopy and infrared spectroscopy showed that the microstructure and secondary structure of collagen changed significantly with the increase in the expansion rate. The penetration of water promotes the rearrangement of hydrogen bonds and ionic bonds between collagen molecules, which not only enhances the interaction between molecules, but also changes the overall conformation of collagen. The mechanism behind these structural changes deserves further discussion. The results of ultraviolet spectroscopy showed that the interaction between collagen molecules increased with the infiltration and distribution of water after the water absorption and expansion of chicken claws. The results of the circular dichroism showed that the expansion treatment resulted in partial collagen denaturation or the partial triple helix structure becoming more relaxed or unwinding, providing more binding space and channels for water molecules, thereby enhancing the water absorption capacity of boneless chicken claws. The results of surface hydrophobicity showed that the expansion treatment of boneless chicken claws changed the tertiary structure of collagen. With the increase in expansion rate, the degree of collagen denaturation gradually increased, and the collagen molecules first stretched and then aggregated, which helped the boneless chicken claws to absorb water more effectively.

Based on the results of water migration, it can be concluded that as the expansion continues, the structural changes of collagen deepen, and the water retention and water absorption properties of collagen are also enhanced. The 50% expansion rate sample had the fastest moisture increase rate, the highest peak value, and the slowest decline rate, showing the best water retention performance. This may be due to its higher water content and larger intercellular space and collagen network gap, which can more effectively maintain free water. In the actual processing and application of chicken claws, the quality of chicken claws is mainly determined by their texture, and the water retention performance is an important indicator that affects the texture of chicken claws, so this discovery is particularly important for the processing of chicken claw products. At the same time, the appropriate expansion rate can significantly improve the water retention rate of the product, improve the taste and flavor. In addition, this study provides some explanations for the function of collagen by studying the changes of collagen in boneless chicken claws under different expansion rates. It has important practical application value for the processing technology of boneless chicken claws. It is helpful to improve the water absorption and water retention performance and taste of chicken claws, and also has certain reference value for the development of high value-added chicken claw products. At the same time, the effects of other factors on the structural changes of collagen have not been involved in this study, and future research can be further explored, such as the effects of temperature and pH on the expansion process. In general, the food industry can refer to this result and select the appropriate expansion treatment process to improve the quality of boneless chicken claws or other similar products, which can further give chicken claw products better texture, flavor and nutritional characteristics, thus attracting a wider range of consumer groups.

## Figures and Tables

**Figure 1 foods-13-03682-f001:**
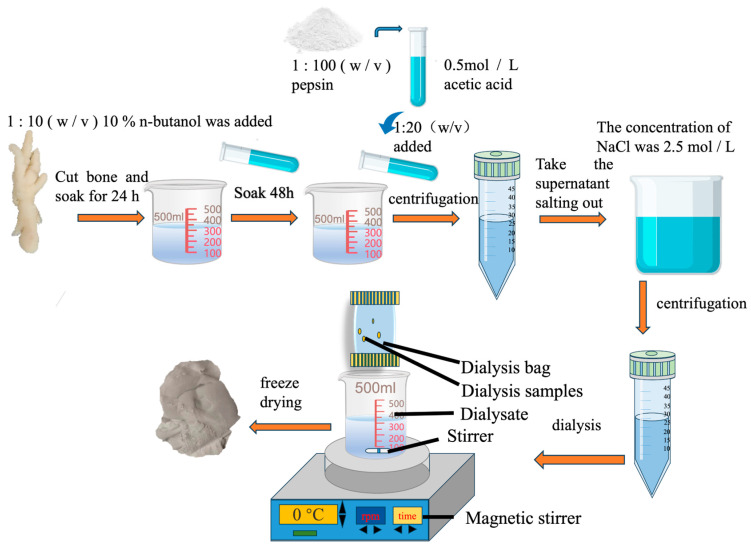
Collagen extraction process.

**Figure 2 foods-13-03682-f002:**
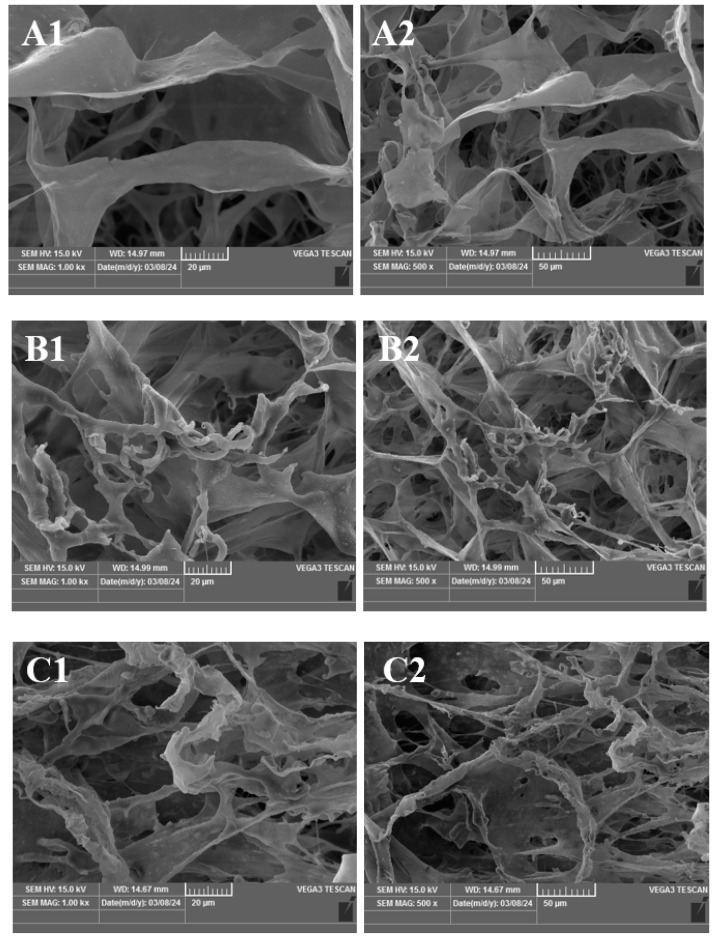
Scanning electron microscope images of chicken claw collagen under different expansion rates. Note: (**A1**–**F1**): Boneless chicken claw collagen (×1000); (**A2**–**F2**): boneless chicken claw collagen (×500); among them, (**A1**,**A2**): 0%; (**B1**,**B2**): 10%; (**C1**,**C2**): 20%; (**D1**,**D2**): 30%; (**E1**,**E2**): 40%; (**F1**,**F2**): 50%.

**Figure 3 foods-13-03682-f003:**
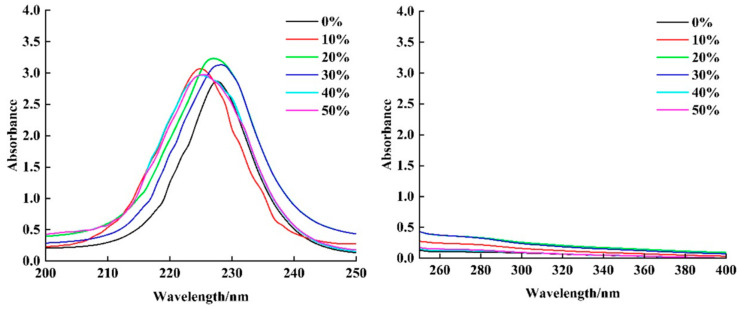
UV–visible spectra of collagen under different expansion rates.

**Figure 4 foods-13-03682-f004:**
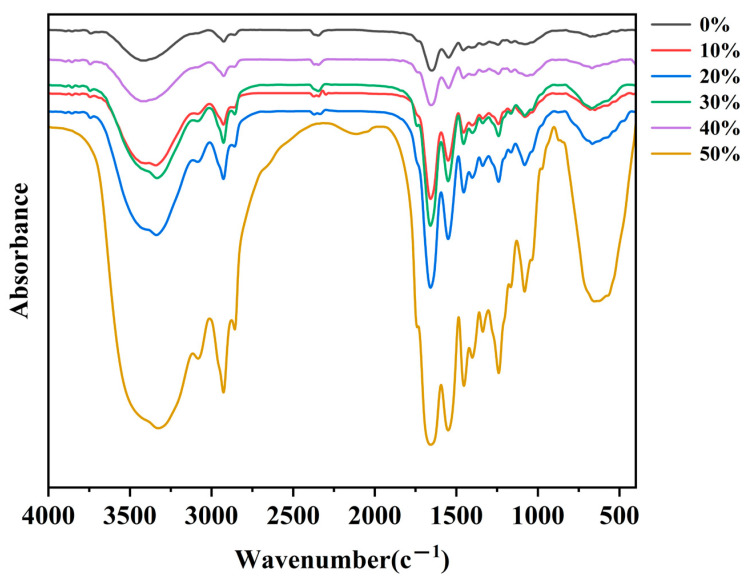
Infrared spectra of collagen under different expansion rates.

**Figure 5 foods-13-03682-f005:**
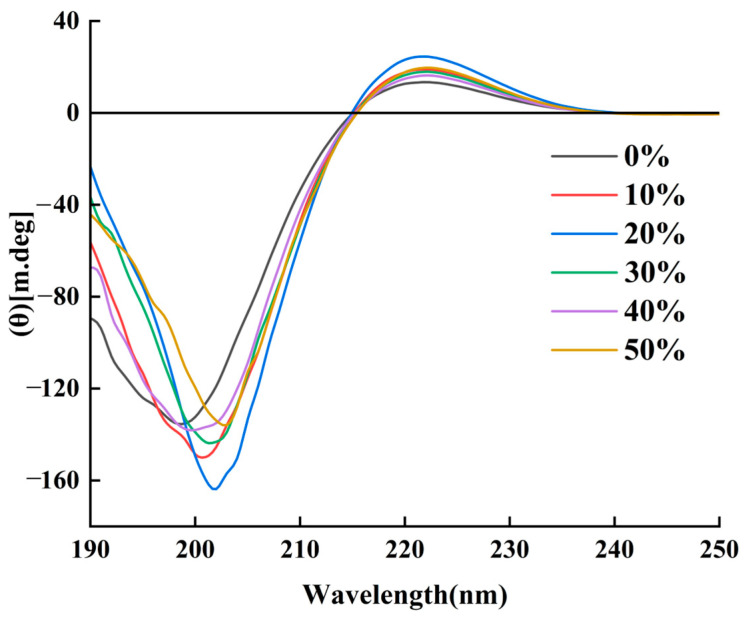
CD spectra of collagen under different expansion rates.

**Figure 6 foods-13-03682-f006:**
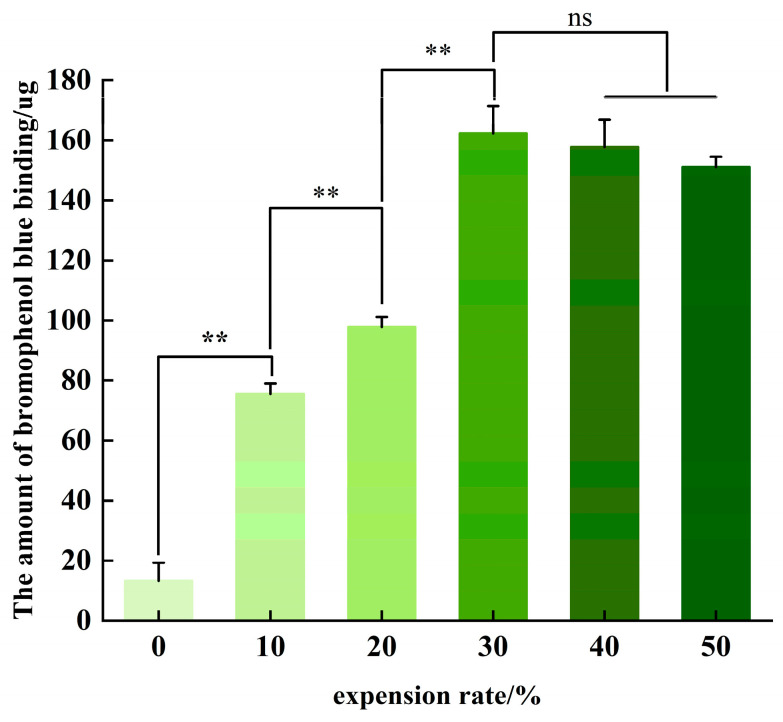
Surface hydrophobicity of collagen under different expansion rates. Note: ** means extremely significant (*p* < 0.01), ns means not significant (*p* > 0.05).

**Figure 7 foods-13-03682-f007:**
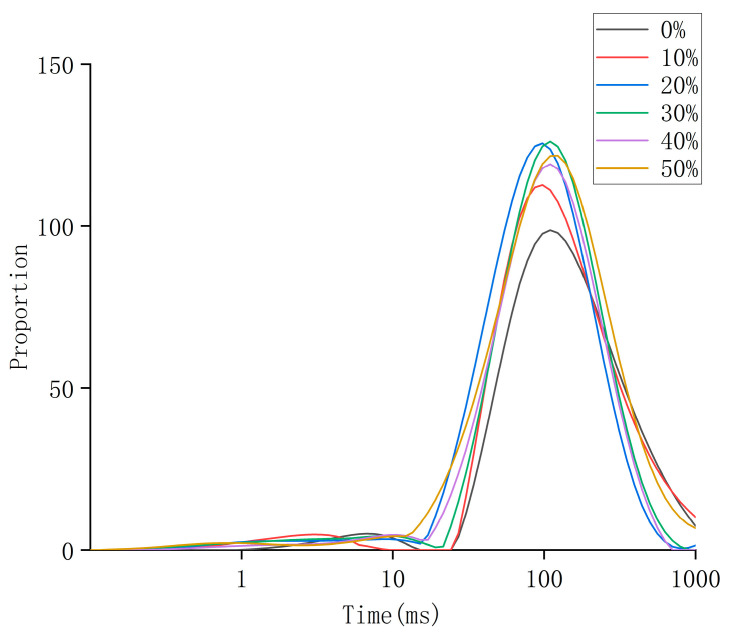
Moisture migration of chicken claws under different expansion rates.

**Table 1 foods-13-03682-t001:** Infrared spectral data of collagen with different expansion rates.

Wave Crest/cm^−1^	0%	10%	20%	30%	40%	50%
AmideA	3417	3343	3341	3345	3418	3325
AmideB	2924	2928	2928	2927	2926	2925
Amide I	1660	1658	1658	1659	1657	1657
Amide II	1548	1550	1550	1550	1547	1549
Amide III	1244	1242	1241	1241	1243	1240

## Data Availability

The original contributions presented in the study are included in the article, further inquiries can be directed to the corresponding author.

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
