# Peer review of "Study on the Structural Changes of Boneless Chicken Claw Collagen and Its Effect on Water Retention Performance"

_foods, 2024, doi:10.3390/foods13223682_

Round 1
Reviewer 1 Report
Comments and Suggestions for Authors
The paper of Tang et al. describes structural changes of collagen derived from boneless chicken claw at different swelling levels and correlates these results with water retention properties. The topic is of interest in the context of the Chinese economy since chicken claws are used as food but they can also be a source of collagen that can be processed and find different applications as in medicine and material science. Giving the industrial interest of collagen from chicken claw, this reviewer considers that this paper is suitable for publication in Foods journal. However, before publication minor points should be addressed and are outlined in the following:
Line 115-116: “according to 1:10” at the beginning of this sentence is not clear
Line 132: Authors should explain why they choose to perform structural analyses of collagen in acetic acid solution
Figure 3-1: it may be useful for the reader to report the % of expansion rate near the images
Line 206: “Ultraviolet Spectroscope Analysis”, better to use spectroscopy.
Line 218: before interpreting the spectra, authors should explain briefly how changes in the maximum of absorption and intensity of the peak at around 230 nm can be correlated to structural changes
Line 227: the title of the paragraph “Infrared spectroscopy” should be added
Line 460: this last sentence of the conclusion should be expanded suggesting an example on how the reported results can be useful for the "development of high value-added chicken claw products".
Comments on the Quality of English Language
The English language of the text is fine and only minor editing is required.
Author Response
Dear Reviewer,
Thank you for your thorough review and valuable suggestions on our manuscript. We have carefully reviewed and addressed each of your comments, with all modifications clearly marked in the text.
Each suggested change is marked in red font, allowing you to quickly locate specific adjustments.
Thank you again for your time and effort in helping us improve the quality of our paper. We are more than willing to make further revisions should you have additional feedback.
Sincerely,
Zheng Tang

Reviewer 2 Report
Comments and Suggestions for Authors
The paper presents a thorough study on the structural changes of chicken claw collagen and its effect on water retention performance, making a significant contribution to the understanding of water retention in meat processing. It uses a variety of appropriate techniques (SEM, UV, FTIR, CD, LF-NMR) to analyze structural changes, which adds to the robustness of the study. However, the manuscript can be improve its quality.
Suggestions for Improvement:
- The abstract should include clearer conclusions about how the findings could impact food science applications, especially regarding product optimization and economic significance, instead of leaving this information for the full paper.
- The introduction, more background on the importance of collagen water retention in food processing, specifically in chicken-based products, could enhance context.
- The introduction could also more clearly outline the novelty of the study. Is this the first time the expansion rates of chicken claw collagen have been studied? More emphasis on the gap in current knowledge would strengthen the paper.
- The discussion section tends to repeat a lot of what is already stated in the results. While the results are comprehensive, the key takeaways should be synthesized more effectively. There is a lack of critical analysis about why certain structural changes occur and what the implications of these changes are.
- The discussion could also link back to broader implications in food science more explicitly. How do these findings translate to industrial applications? Are there practical recommendations for optimizing water retention in processed chicken products?
- The conclusion could benefit from being more concise. Repeating detailed experimental results (e.g., absorption peaks, expansion rates) in the conclusion is unnecessary.
- A more future-oriented approach in the conclusion would be valuable. The authors could discuss potential future studies, such as exploring other variables like temperature, pH, or further refinement of the expansion process for different food products.
- There is a degree of repetition, particularly in the results and conclusion sections. A more succinct and refined writing style would be beneficial.
Comments on the Quality of English Language
The technical language used throughout the paper is appropriate for a scientific audience.
Author Response
Dear Reviewer,
Thank you for your thorough review and valuable suggestions on our manuscript. We have carefully reviewed and addressed each of your comments, with all modifications clearly marked in the text.
Each suggested change is marked in green font, allowing you to quickly locate specific adjustments.
Thank you again for your time and effort in helping us improve the quality of our paper. We are more than willing to make further revisions should you have additional feedback.
Sincerely,
Zheng Tang

Reviewer 3 Report
Comments and Suggestions for Authors
Dear author, the information contained in this manuscript is interesting, however, I consider it appropriate to make some changes among them:
Summary section
I consider it appropriate to include in the initial information section the economic importance of chicken claw products as described in the introduction section.
1. Introduction section
Line 85 describes that there are studies on collagen and could include more information.
It could indicate why water retention after swelling is a crucial indicator in chicken claw products (line 87).
2.- Materials section
It is suggested that specific information on the acquisition of chicken claw be included, and that the wording of the information contained in lines 92-94 be modified, since the current wording is interpreted as storage instructions for the product and not as an activity carried out in the research.
2.1.3. Instruments and Equipment
It is recommended that the instruments described in this section be included in each determination in which they were used and not be presented as an isolated section.
2.2.1. Preparation of Boneless Chicken Claw at Different Expansion Rates
-I consider it important to modify the wording of this information to obtain an adequate description of the method followed “ Consult the method of Zheng Qingyao[xii] et al. to extract collagen from sea cucumber, and modify it according”.
-Could include information on the “different expansion cups” they consisted of (line 114).
2.2.2 Determination of Scanning Electron Microscopy
It is recommended to include information on the microscope used
- It is recommended to include in the first figure presented a figure caption describing what it consists of, as well as to include it in the text that writes it.
2.2.4. Determination by Fourier Infrared Spectroscopy
Include the brand of the equipment used
2.2.5. Determination of Circular Dichroism
Include mark of equipment used
2.2.6. Determination of Surface Hydrophobicity
2.2.7. LF-NMR Moisture Migration Determination
- It is necessary to include characteristics of the equipment used
Results and discussion section
-It is recommended to include only one number to cite the figures, as well as in the figure footnote numbering.
- It is important to review the figure numbering and make sure that it agrees with the text that describes them.
- It is also important that all the figure captions describe the figure completely.
3.2. Ultraviolet Spectroscope Analysis
- Lines 209-210 describe aromatic amino acids of collagen type i, could you include information on what they are?
- it is recommended that the figure be set as “Figure 3-2. UV-visible spectra of collagen under different expansion rates”, it is recommended that the figure be included in (a) and (b) to describe the 2 images presented.
- In line 227 the results of Fig. 3-3 are described, however this is not presented, I consider that there is a confusion and it corresponds to figure 3.4, could you check?
3.3. Circular Dichroism Analysis
It is important to revise the enumeration of figures
I also consider it appropriate to include the complete word Circular Dichroism instead of CD of the figure described as 3.4.
3.4. Surface Hydrophobicity Analysis
It is relevant to include the data of the figure in the English language (X and Y axis information), as well as in the figure caption.
3.5. LF-NMR Analysis
The figure described as 3.6 is not related in the text, also it is considered appropriate to include more information that describes the figure “Water migration of collagen”, that is to say to include information of where is the collagen?
Conclusion section
It is important to restructure this section, the conclusion of the research should be short and concise. Also, do not include a relationship between the conclusion and the objective of the research.
According to the content of this section, it represents a summary of the results and discussion of them, so I suggest that it be modified. It is possible to include information that is not duplicated as part of the results and discussion section described above.
References section
It is considered appropriate to carry out an exhaustive revision and correction of the references described, it is necessary to describe them completely and in the format established by the journal, as well as to cite them adequately in the text of the manuscript.
Author Response
Dear Reviewer,
Thank you for your thorough review and valuable suggestions on our manuscript. We have carefully reviewed and addressed each of your comments, with all modifications clearly marked in the text.
Responses are marked in blank font, allowing you to quickly locate specific adjustments.
Thank you again for your time and effort in helping us improve the quality of our paper. We are more than willing to make further revisions should you have additional feedback.
Sincerely,
Zheng Tang

Round 2
Reviewer 3 Report
Comments and Suggestions for Authors
Dear authors, I thank you for having considered the observations made to the manuscript, however, I consider that it is still important to modify some other aspects that were not modified, among these:
1.- It is recommended to revise the enumeration of the figures, because it is not correct, for example, there are two figures that have the enumeration 3-4, and 3-3 is not found.
2.-It is recommended to review that the figures are described in the text, for example figure 3.6 is not described or related in the text.
3.-It is also recommended to modify the conclusion section as previously requested, this should be short and concrete and not contain information that can be included as part of the results or their discussions.
4.- An exhaustive revision of the bibliography is requested in order to ensure that it is in the format requested by the journal.
Author Response
Dear Reviewer, I have revised the manuscript according to the suggested requirements, and the revisions are marked based on which specific suggestions.
Please see the attachment.
